# SEARCHING FOR ACTIVATION FUNCTIONS

## ABSTRACT

The choice of activation functions in deep networks has a significant effect on the training dynamics and task performance. Currently, the most successful and widely-used activation function is the Rectified Linear Unit (ReLU). Although various hand-designed alternatives to ReLU have been proposed, none have managed to replace it due to inconsistent gains. In this work, we propose to leverage automatic search techniques to discover new activation functions. Using a combination of exhaustive and reinforcement learning-based search, we discover multiple novel activation functions. We verify the effectiveness of the searches by conducting an empirical evaluation with the best discovered activation function. Our experiments show that the best discovered activation function, $f(x) = x \cdot \text{sigmoid}(\beta x)$, which we name Swish, tends to work better than ReLU on deeper models across a number of challenging datasets. For example, simply replacing ReLUs with Swish units improves top-1 classification accuracy on ImageNet by 0.9% for Mobile NASNet-A and 0.6% for Inception-ResNet-v2. The simplicity of Swish and its similarity to ReLU make it easy for practitioners to replace ReLUs with Swish units in any neural network.

## 1 INTRODUCTION

At the heart of every deep network lies a linear transformation followed by an activation function $f(\cdot)$. The activation function plays a major role in the success of training deep neural networks. Currently, the most successful and widely-used activation function is the Rectified Linear Unit (ReLU) (Hahnloser et al., 2000; Jarrett et al., 2009; Nair & Hinton, 2010), defined as $f(x) = \max(x, 0)$. The use of ReLUs was a breakthrough that enabled the fully supervised training of state-of-the-art deep networks (Krizhevsky et al., 2012). Deep networks with ReLUs are more easily optimized than networks with sigmoid or tanh units, because gradients are able to flow when the input to the ReLU function is positive. Thanks to its simplicity and effectiveness, ReLU has become the default activation function used across the deep learning community.

While numerous activation functions have been proposed to replace ReLU (Maas et al., 2013; He et al., 2015; Clevert et al., 2015; Klambauer et al., 2017), none have managed to gain the widespread adoption that ReLU enjoys. Many practitioners have favored the simplicity and reliability of ReLU because the performance improvements of the other activation functions tend to be inconsistent across different models and datasets.

The activation functions proposed to replace ReLU were hand-designed to fit properties deemed to be important. However, the use of search techniques to automate the discovery of traditionally human-designed components has recently shown to be extremely effective (Zoph & Le, 2016; Bello et al., 2017; Zoph et al., 2017). For example, Zoph et al. (2017) used reinforcement learning-based search to find a replicable convolutional cell that outperforms human-designed architectures on ImageNet.

In this work, we use automated search techniques to discover novel activation functions. We focus on finding new scalar activation functions, which take in as input a scalar and output a scalar, because scalar activation functions can be used to replace the ReLU function without changing the network architecture. Using a combination of exhaustive and reinforcement learning-based search, we find a number of novel activation functions that show promising performance. To further validate the effectiveness of using searches to discover scalar activation functions, we empirically evaluate the best discovered activation function. The best discovered activation function, which we call *Swish*, is

$f(x) = x \cdot \text{sigmoid}(\beta x)$, where $\beta$ is a constant or trainable parameter. Our extensive experiments show that Swish consistently matches or outperforms ReLU on deep networks applied to a variety of challenging domains such as image classification and machine translation. On ImageNet, replacing ReLUs with Swish units improves top-1 classification accuracy by 0.9% on Mobile NASNet-A (Zoph et al., 2017) and 0.6% on Inception-ResNet-v2 (Szegedy et al., 2017). These accuracy gains are significant given that one year of architectural tuning and enlarging yielded 1.3% accuracy improvement going from Inception V3 (Szegedy et al., 2016) to Inception-ResNet-v2 (Szegedy et al., 2017).

## 2 METHODS

In order to utilize search techniques, a search space that contains promising candidate activation functions must be designed. An important challenge in designing search spaces is balancing the size and expressivity of the search space. An overly constrained search space will not contain novel activation functions, whereas a search space that is too large will be difficult to effectively search. To balance the two criteria, we design a simple search space inspired by the optimizer search space of Bello et al. (2017) that composes unary and binary functions to construct the activation function.

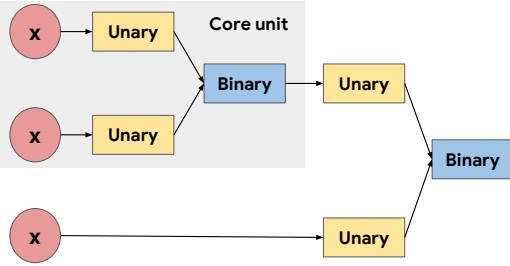

Figure 1: An example activation function structure. The activation function is composed of multiple repetitions of the "core unit", which consists of two inputs, two unary functions, and one binary function. Unary functions take in a single scalar input and return a single scalar output, such $u(x) = x^2$ or $u(x) = \sigma(x)$. Binary functions take in two scalar inputs and return a single scalar output, such as $b(x_1, x_2) = x_1 \cdot x_2$ or $b(x_1, x_2) = \exp(-(x_1 - x_2)^2)$.

As shown in Figure 1, the activation function is constructed by repeatedly composing the the "core unit", which is defined as $b(u_1(x_1), u_2(x_2))$. The core unit takes in two scalar inputs, passes each input independently through an unary function, and combines the two unary outputs with a binary function that outputs a scalar. Since our aim is to find scalar activation functions which transform a single scalar input into a single scalar output, the inputs of the unary functions are restricted to the layer preactivation $x$ and the binary function outputs.

Given the search space, the goal of the search algorithm is to find effective choices for the unary and binary functions. The choice of the search algorithm depends on the size of the search space. If the search space is small, such as when using a single core unit, it is possible to exhaustively enumerate the entire search space. If the core unit is repeated multiple times, the search space will be extremely large (i.e., on the order of $10^{12}$ possibilities), making exhaustive search infeasible.

For large search spaces, we use an RNN controller (Zoph & Le, 2016), which is visualized in Figure 2. At each timestep, the controller predicts a single component of the activation function. The prediction is fed back to the controller in the next timestep, and this process is repeated until every component of the activation function is predicted. The predicted string is then used to construct the activation function.

Once a candidate activation function has been generated by the search algorithm, a "child network" with the candidate activation function is trained on some task, such as image classification on CIFAR-10. After training, the validation accuracy of the child network is recorded and used to update the search algorithm. In the case of exhaustive search, a list of the top performing activation functions ordered by validation accuracy is maintained. In the case of the RNN controller,

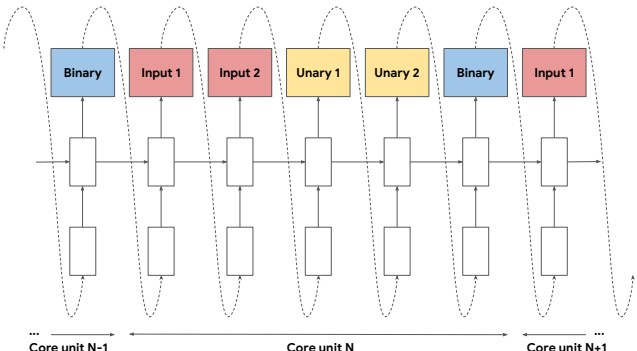

Figure 2: The RNN controller used to search over large spaces. At each step, it predicts a single component of the activation function. The prediction is fed back as input to the next timestep in an autoregressive fashion. The controller keeps predicting until every component of the activation function has been chosen. The controller is trained with reinforcement learning.

the controller is trained with reinforcement learning to maximize the validation accuracy, where the validation accuracy serves as the reward. This training pushes the controller to generate activation functions that have high validation accuracies.

Since evaluating a single activation function requires training a child network, the search is computationally expensive. To decrease the wall clock time required to conduct the search, a distributed training scheme is used to parallelize the training of each child network. In this scheme, the search algorithm proposes a batch of candidate activation functions which are added to a queue. Worker machines pull activation functions off the queue, train a child network, and report back the final validation accuracy of the corresponding activation function. The validation accuracies are aggregated and used to update the search algorithm.

## 3 SEARCH FINDINGS

We conduct all our searches with the ResNet-20 (He et al., 2016a) as the child network architecture, and train on CIFAR-10 (Krizhevsky & Hinton, 2009) for 10K steps. This constrained environment could potentially skew the results because the top performing activation functions might only perform well for small networks. However, we show in the experiments section that many of the discovered functions generalize to larger models. Exhaustive search is used for small search spaces, while an RNN controller is used for larger search spaces. The RNN controller is trained with Proximal Policy Optimization (Schulman et al., 2017), using the exponential moving average of rewards as a baseline to reduce variance. The full list unary and binary functions considered are as follows:

- **Unary functions**: $x, -x, |x|, x^2, x^3, \sqrt{x}, \beta x, x + \beta, \log(|x| + \epsilon), \exp(x) \sin(x), \cos(x), \sinh(x), \cosh(x), \tanh(x), \sinh^{-1}(x), \tan^{-1}(x), \text{sinc}(x), \max(x, 0), \min(x, 0), \sigma(x), \log(1 + \exp(x)), \exp(-x^2), \text{erf}(x), \beta$

- **Binary functions**: $x_1 + x_2, x_1 \cdot x_2, x_1 - x_2, \frac{x_1}{x_2 + \epsilon}, \max(x_1, x_2), \min(x_1, x_2), \sigma(x_1) \cdot x_2, \exp(-\beta(x_1 - x_2)^2), \exp(-\beta|x_1 - x_2|), \beta x_1 + (1 - \beta)x_2$

where $\beta$ indicates a per-channel trainable parameter and $\sigma(x) = (1 + \exp(-x))^{-1}$ is the sigmoid function. Different search spaces are created by varying the number of core units used to construct the activation function and varying the unary and binary functions available to the search algorithm.

Figure 3 plots the top performing novel activation functions found by the searches. We highlight several noteworthy trends uncovered by the searches:

- Complicated activation functions consistently underperform simpler activation functions, potentially due to an increased difficulty in optimization. The best performing activation functions can be represented by 1 or 2 core units.

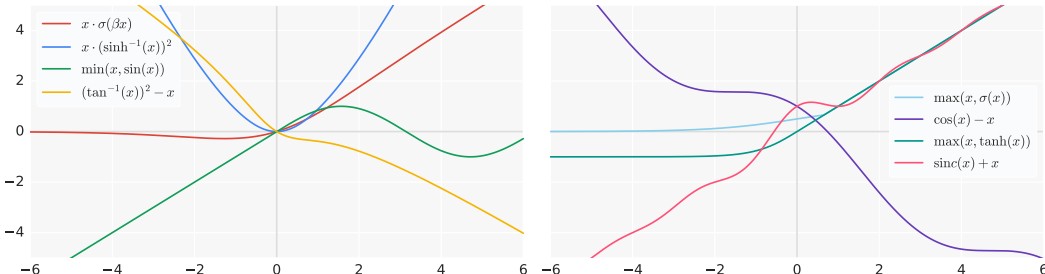

Figure 3: The top novel activation functions found by the searches. Separated into two diagrams for visual clarity. Best viewed in color.

- A common structure shared by the top activation functions is the use of the raw preactivation $x$ as input to the final binary function: $b(x, g(x))$. The ReLU function also follows this structure, where $b(x_1, x_2) = \max(x_1, x_2)$ and $g(x) = 0$.

- The searches discovered activation functions that utilize periodic functions, such as $\sin$ and $\cos$. The most common use of periodic functions is through addition or subtraction with the raw preactivation $x$ (or a linearly scaled $x$). The use of periodic functions in activation functions has only been briefly explored in prior work (Parascandolo et al., 2016), so these discovered functions suggest a fruitful route for further research.

- Functions that use division tend to perform poorly because the output explodes when the denominator is near $0$. Division is successful only when functions in the denominator are either bounded away from $0$, such as $\cosh(x)$, or approach $0$ only when the numerator also approaches $0$, producing an output of $1$.

Since the activation functions were found using a relatively small child network, their performance may not generalize when applied to bigger models. To test the robustness of the top performing novel activation functions to different architectures, we run additional experiments using the preactivation ResNet-164 (RN) (He et al., 2016b), Wide ResNet 28-10 (WRN) (Zagoruyko & Komodakis, 2016), and DenseNet 100-12 (DN) (Huang et al., 2017) models. We implement the 3 models in TensorFlow and replace the ReLU function with each of the top novel activation functions discovered by the searches. We use the same hyperparameters described in each work, such as optimizing using SGD with momentum, and follow previous works by reporting the median of 5 different runs.

| Function | RN | WRN | DN |
|---|---|---|---|
| ReLU $[\max(x, 0)]$ | 93.8 | 95.3 | 94.8 |
| $x \cdot \sigma(\beta x)$ | 94.5 | 95.5 | 94.9 |
| $\max(x, \sigma(x))$ | 94.3 | 95.3 | 94.8 |
| $\cos(x) - x$ | 94.1 | 94.8 | 94.6 |
| $\min(x, \sin(x))$ | 94.0 | 95.1 | 94.4 |
| $(\tan^{-1}(x))^2 - x$ | 93.9 | 94.7 | 94.9 |
| $\max(x, \tanh(x))$ | 93.9 | 94.2 | 94.5 |
| $\mathrm{sinc}(x) + x$ | 91.5 | 92.1 | 92.0 |
| $x \cdot (\sinh^{-1}(x))^2$ | 85.1 | 92.1 | 91.1 |

Table 1: CIFAR-10 accuracy.

| Function | RN | WRN | DN |
|---|---|---|---|
| ReLU $[\max(x, 0)]$ | 74.2 | 77.8 | 83.7 |
| $x \cdot \sigma(\beta x)$ | 75.1 | 78.0 | 83.9 |
| $\max(x, \sigma(x))$ | 74.8 | 78.6 | 84.2 |
| $\cos(x) - x$ | 75.2 | 76.6 | 81.8 |
| $\min(x, \sin(x))$ | 73.4 | 77.1 | 74.3 |
| $(\tan^{-1}(x))^2 - x$ | 75.2 | 76.7 | 83.1 |
| $\max(x, \tanh(x))$ | 74.8 | 76.0 | 78.6 |
| $\mathrm{sinc}(x) + x$ | 66.1 | 68.3 | 67.9 |
| $x \cdot (\sinh^{-1}(x))^2$ | 52.8 | 70.6 | 68.1 |

Table 2: CIFAR-100 accuracy.

The results are shown in Tables 1 and 2. Despite the changes in model architecture, six of the eight activation functions successfully generalize. Of these six activation functions, all match or outperform ReLU on ResNet-164. Furthermore, two of the discovered activation functions, $x \cdot \sigma(\beta x)$ and $\max(x, \sigma(x))$, consistently match or outperform ReLU on all three models.

While these results are promising, it is still unclear whether the discovered activation functions can successfully replace ReLU on challenging real world datasets. In order to validate the effectiveness of the searches, in the rest of this work we focus on empirically evaluating the activation function $f(x) = x \cdot \sigma(\beta x)$, which we call *Swish*. We choose to extensively evaluate Swish in-

stead of $\max(x, \sigma(x))$ because early experimentation showed better generalization for Swish. In the following sections, we analyze the properties of Swish and then conduct a thorough empirical evaluation comparing Swish, ReLU, and other candidate baseline activation functions on number of large models across a variety of tasks.

## 4 SWISH

To recap, Swish is defined as $x \cdot \sigma(\beta x)$, where $\sigma(z) = (1 + \exp(-z))^{-1}$ is the sigmoid function and $\beta$ is either a constant or a trainable parameter. Figure 4 plots the graph of Swish for different values of $\beta$. If $\beta = 1$, Swish is equivalent to the Sigmoid-weighted Linear Unit (SiL) of Elfwing et al. (2017) that was proposed for reinforcement learning. If $\beta = 0$, Swish becomes the scaled linear function $f(x) = \frac{x}{2}$. As $\beta \to \infty$, the sigmoid component approaches a 0-1 function, so Swish becomes like the ReLU function. This suggests that Swish can be loosely viewed as a smooth function which nonlinearly interpolates between the linear function and the ReLU function. The degree of interpolation can be controlled by the model if $\beta$ is set as a trainable parameter.

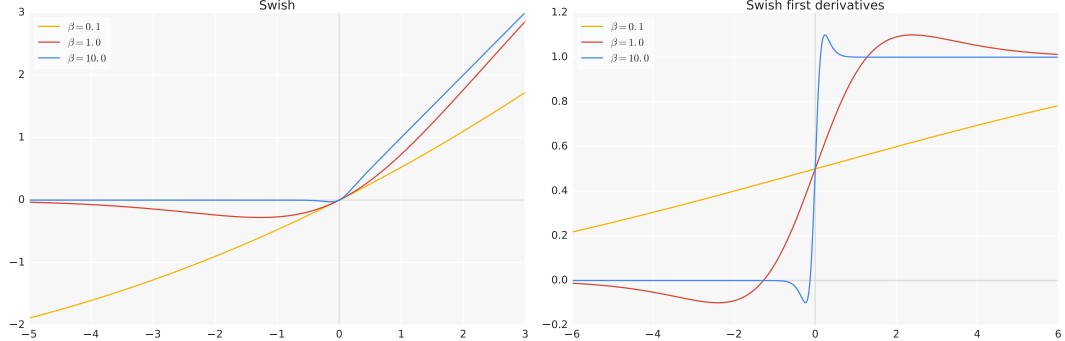

Figure 4: The Swish activation function.   Figure 5: First derivatives of Swish.

Like ReLU, Swish is unbounded above and bounded below. Unlike ReLU, Swish is smooth and non-monotonic. In fact, the non-monotonicity property of Swish distinguishes itself from most common activation functions. The derivative of Swish is

$$
\begin{aligned}
f'(x) &= \sigma(\beta x) + \beta x \cdot \sigma(\beta x)(1 - \sigma(\beta x)) \\
&= \sigma(\beta x) + \beta x \cdot \sigma(\beta x) - \beta x \cdot \sigma(\beta x)^2 \\
&= \beta x \cdot \sigma(x) + \sigma(\beta x)(1 - \beta x \cdot \sigma(\beta x)) \\
&= \beta f(x) + \sigma(\beta x)(1 - \beta f(x))
\end{aligned}
$$

The first derivative of Swish is shown in Figure 5 for different values of $\beta$. The scale of $\beta$ controls how fast the first derivative asymptotes to 0 and 1. When $\beta = 1$, the derivative has magnitude less than 1 for inputs that are less than around 1.25. Thus, the success of Swish with $\beta = 1$ implies that the gradient preserving property of ReLU (i.e., having a derivative of 1 when $x > 0$) may no longer be a distinct advantage in modern architectures.

The most striking difference between Swish and ReLU is the non-monotonic "bump" of Swish when $x < 0$. As shown in Figure 6, a large percentage of preactivations fall inside the domain of the bump ($-5 \le x \le 0$), which indicates that the non-monotonic bump is an important aspect of Swish. The shape of the bump can be controlled by changing the $\beta$ parameter. While fixing $\beta = 1$ is effective in practice, the experiments section shows that training $\beta$ can further improve performance on some models. Figure 7 plots distribution of trained $\beta$ values from a Mobile NASNet-A model (Zoph et al., 2017). The trained $\beta$ values are spread out between 0 and 1.5 and have a peak at $\beta \approx 1$, suggesting that the model takes advantage of the additional flexibility of trainable $\beta$ parameters.

Practically, Swish can be implemented with a single line code change in most deep learning libraries, such as TensorFlow (Abadi et al., 2016) (e.g., `x * tf.sigmoid(beta * x)` or `tf.nn.swish(x)` if using a version of TensorFlow released after the submission of this work). As a cautionary note, if BatchNorm (Ioffe & Szegedy, 2015) is used, the scale parameter should be

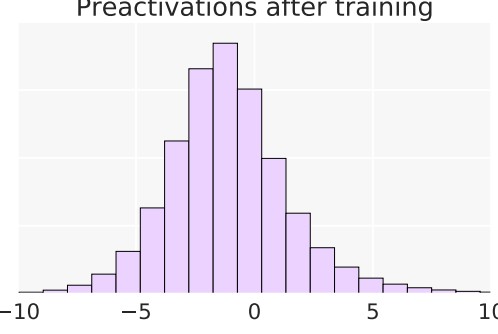

Figure 6: Preactivation distribution after training of Swish with $\beta = 1$ on ResNet-32.

Figure 7: Distribution of trained $\beta$ values of Swish on Mobile NASNet-A.

set. Some high level libraries turn off the scale parameter by default due to the ReLU function being piecewise linear, but this setting is incorrect for Swish. For training Swish networks, we found that slightly lowering the learning rate used to train ReLU networks works well.

## 5 EXPERIMENTS WITH SWISH

We benchmark Swish against ReLU and a number of recently proposed activation functions on challenging datasets, and find that Swish matches or exceeds the baselines on nearly all tasks. The following sections will describe our experimental settings and results in greater detail. As a summary, Table 3 shows Swish in comparison to each baseline activation function we considered (which are defined in the next section). The results in Table 3 are aggregated by comparing the performance of Swish to the performance of different activation functions applied to a variety of models, such as Inception ResNet-v2 (Szegedy et al., 2017) and Transformer (Vaswani et al., 2017), across multiple datasets, such as CIFAR, ImageNet, and English→German translation.[1] The improvement of Swish over other activation functions is statistically significant under a one-sided paired sign test.

| Baselines | ReLU | LReLU | PReLU | Softplus | ELU | SELU | GELU |
|---|---|---|---|---|---|---|---|
| Swish > Baseline | 9 | 7 | 6 | 6 | 8 | 8 | 8 |
| Swish = Baseline | 0 | 1 | 3 | 2 | 0 | 1 | 1 |
| Swish < Baseline | 0 | 1 | 0 | 1 | 1 | 0 | 0 |

Table 3: The number of models on which Swish outperforms, is equivalent to, or underperforms each baseline activation function we compared against in our experiments.

### 5.1 EXPERIMENTAL SET UP

We compare Swish against several additional baseline activation functions on a variety of models and datasets. Since many activation functions have been proposed, we choose the most common activation functions to compare against, and follow the guidelines laid out in each work:

- Leaky ReLU (*LReLU*) (Maas et al., 2013):

$$f(x) = \begin{cases} x & \text{if } x \geq 0 \\ \alpha x & \text{if } x < 0 \end{cases}$$

where $\alpha = 0.01$. LReLU enables a small amount of information to flow when $x < 0$.

---

[1]To avoid skewing the comparison, each model type is compared just once. A model with multiple results is represented by the median of its results. Specifically, the models with aggregated results are (a) ResNet-164, Wide ResNet 28-10, and DenseNet 100-12 across the CIFAR-10 and CIFAR-100 results, (b) Mobile NASNet-A and Inception-ResNet-v2 across the 3 runs, and (c) WMT Transformer model across the 4 newstest results.

- Parametric ReLU (*PReLU*) (He et al., 2015): The same form as LReLU but $\alpha$ is a learnable parameter. Each channel has a shared $\alpha$ which is initialized to $0.25$.

- Softplus (Nair & Hinton, 2010): $f(x) = \log(1 + \exp(x))$. Softplus is a smooth function with properties similar to Swish, but is strictly positive and monotonic. It can be viewed as a smooth version of ReLU.

- Exponential Linear Unit (*ELU*) (Clevert et al., 2015):

$$f(x) = \begin{cases} x & \text{if } x \geq 0 \\ \alpha(\exp(x) - 1) & \text{if } x < 0 \end{cases}$$

where $\alpha = 1.0$

- Scaled Exponential Linear Unit (*SELU*) (Klambauer et al., 2017):

$$f(x) = \lambda \begin{cases} x & \text{if } x \geq 0 \\ \alpha(\exp(x) - 1) & \text{if } x < 0 \end{cases}$$

with $\alpha \approx 1.6733$ and $\lambda \approx 1.0507$.

- Gaussian Error Linear Unit (*GELU*) (Hendrycks & Gimpel, 2016): $f(x) = x \cdot \Phi(x)$, where $\Phi(x)$ is the cumulative distribution function of the standard normal distribution. GELU is a nonmonotonic function that has a shape similar to Swish with $\beta = 1.4$.

We evaluate both Swish with a trainable $\beta$ and Swish with a fixed $\beta = 1$ (which for simplicity we call Swish-1, but it is equivalent to the Sigmoid-weighted Linear Unit of Elfwing et al. (2017)). Note that our results may not be directly comparable to the results in the corresponding works due to differences in our training setup.

## 5.2 CIFAR

We first compare Swish to all the baseline activation functions on the CIFAR-10 and CIFAR-100 datasets (Krizhevsky & Hinton, 2009). We follow the same set up used when comparing the activation functions discovered by the search techniques, and compare the median of 5 runs with the preactivation ResNet-164 (He et al., 2016b), Wide ResNet 28-10 (WRN) (Zagoruyko & Komodakis, 2016), and DenseNet 100-12 (Huang et al., 2017) models.

| Model | ResNet | WRN | DenseNet |
|---|---|---|---|
| LReLU | 94.2 | 95.6 | 94.7 |
| PReLU | 94.1 | 95.1 | 94.5 |
| Softplus | 94.6 | 94.9 | 94.7 |
| ELU | 94.1 | 94.1 | 94.4 |
| SELU | 93.0 | 93.2 | 93.9 |
| GELU | 94.3 | 95.5 | 94.8 |
| ReLU | 93.8 | 95.3 | 94.8 |
| Swish-1 | 94.7 | 95.5 | 94.8 |
| Swish | 94.5 | 95.5 | 94.8 |

Table 4: CIFAR-10 accuracy.

| Model | ResNet | WRN | DenseNet |
|---|---|---|---|
| LReLU | 74.2 | 78.0 | 83.3 |
| PReLU | 74.5 | 77.3 | 81.5 |
| Softplus | 76.0 | 78.4 | 83.7 |
| ELU | 75.0 | 76.0 | 80.6 |
| SELU | 73.2 | 74.3 | 80.8 |
| GELU | 74.7 | 78.0 | 83.8 |
| ReLU | 74.2 | 77.8 | 83.7 |
| Swish-1 | 75.1 | 78.5 | 83.8 |
| Swish | 75.1 | 78.0 | 83.9 |

Table 5: CIFAR-100 accuracy.

The results in Tables 4 and 5 show how Swish and Swish-1 consistently matches or outperforms ReLU on every model for both CIFAR-10 and CIFAR-100. Swish also matches or exceeds the best baseline performance on almost every model. Importantly, the "best baseline" changes between different models, which demonstrates the stability of Swish to match these varying baselines. Softplus, which is smooth and approaches zero on one side, similar to Swish, also has strong performance.

## 5.3 IMAGENET

Next, we benchmark Swish against the baseline activation functions on the ImageNet 2012 classification dataset (Russakovsky et al., 2015). ImageNet is widely considered one of most important

image classification datasets, consisting of a 1,000 classes and 1.28 million training images. We evaluate on the validation dataset, which has 50,000 images.

We compare all the activation functions on a variety of architectures designed for ImageNet: Inception-ResNet-v2, Inception-v4, Inception-v3 (Szegedy et al., 2017), MobileNet (Howard et al., 2017), and Mobile NASNet-A (Zoph et al., 2017). All these architectures were designed with Re-LUs. We again replace the ReLU activation function with different activation functions and train for a fixed number of steps, determined by the convergence of the ReLU baseline. For each activation function, we try 3 different learning rates with RMSProp (Tieleman & Hinton, 2012) and pick the best.[2] All networks are initialized with He initialization (He et al., 2015).[3] To verify that the performance differences are reproducible, we run the Inception-ResNet-v2 and Mobile NASNet-A experiments 3 times with the best learning rate from the first experiment. We plot the learning curves for Mobile NASNet-A in Figure 8.

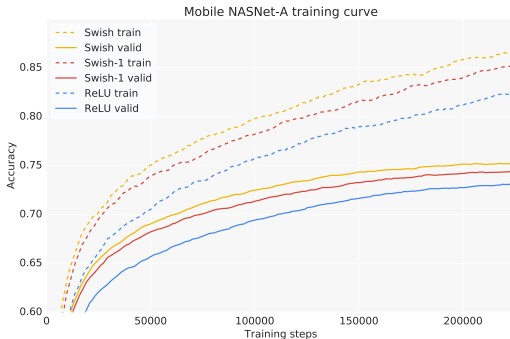

Figure 8: Training curves of Mobile NASNet-A on ImageNet. Best viewed in color

| Model | Top-1 Acc. (%) | | | Top-5 Acc. (%) | | |
|---|---|---|---|---|---|---|
| LReLU | 73.8 | 73.9 | 74.2 | 91.6 | 91.9 | 91.9 |
| PReLU | 74.6 | 74.7 | 74.7 | 92.4 | 92.3 | 92.3 |
| Softplus | 74.0 | 74.2 | 74.2 | 91.6 | 91.8 | 91.9 |
| ELU | 74.1 | 74.2 | 74.2 | 91.8 | 91.8 | 91.8 |
| SELU | 73.6 | 73.7 | 73.7 | 91.6 | 91.7 | 91.7 |
| GELU | 74.6 | - | - | 92.0 | - | - |
| ReLU | 73.5 | 73.6 | 73.8 | 91.4 | 91.5 | 91.6 |
| Swish-1 | 74.6 | 74.7 | 74.7 | 92.1 | 92.0 | 92.0 |
| Swish | 74.9 | 74.9 | 75.2 | 92.3 | 92.4 | 92.4 |

Table 6: Mobile NASNet-A on ImageNet, with 3 different runs ordered by top-1 accuracy. The additional 2 GELU experiments are still training at the time of submission.

| Model | Top-1 Acc. (%) | | | Top-5 Acc. (%) | | |
|---|---|---|---|---|---|---|
| LReLU | 79.5 | 79.5 | 79.6 | 94.7 | 94.7 | 94.7 |
| PReLU | 79.7 | 79.8 | 80.1 | 94.8 | 94.9 | 94.9 |
| Softplus | 80.1 | 80.2 | 80.4 | 95.2 | 95.2 | 95.3 |
| ELU | 75.8 | 79.9 | 80.0 | 92.6 | 95.0 | 95.1 |
| SELU | 79.0 | 79.2 | 79.2 | 94.5 | 94.4 | 94.5 |
| GELU | 79.6 | 79.6 | 79.9 | 94.8 | 94.8 | 94.9 |
| ReLU | 79.5 | 79.6 | 79.8 | 94.8 | 94.8 | 94.8 |
| Swish-1 | 80.2 | 80.3 | 80.4 | 95.1 | 95.2 | 95.2 |
| Swish | 80.2 | 80.2 | 80.3 | 95.0 | 95.2 | 95.0 |

Table 7: Inception-ResNet-v2 on ImageNet with 3 different runs. Note that the ELU sometimes has instabilities at the start of training, which accounts for the first result.

| Model | Top-1 Acc. (%) | Top-5 Acc. (%) |
|---|---|---|
| LReLU | 72.5 | 91.0 |
| PReLU | 74.2 | 91.9 |
| Softplus | 73.6 | 91.6 |
| ELU | 73.9 | 91.3 |
| SELU | 73.2 | 91.0 |
| GELU | 73.5 | 91.4 |
| ReLU | 72.0 | 90.8 |
| Swish-1 | 74.2 | 91.6 |
| Swish | 74.2 | 91.7 |

Table 8: MobileNet on ImageNet.

The results in Tables 6-10 show strong performance for Swish. On Inception-ResNet-v2, Swish outperforms ReLU by a nontrivial 0.5%. Swish performs especially well on mobile sized models, with a 1.4% boost on Mobile NASNet-A and a 2.2% boost on MobileNet over ReLU. Swish also matches or exceeds the best performing baseline on most models, where again, the best performing baseline differs depending on the model. Softplus achieves accuracies comparable to Swish on the

---

[2]For some of the models with ELU, SELU, and PReLU, we train with an additional 3 learning rates (so a total of 6 learning rates) because the original 3 learning rates did not converge.

[3]For SELU, we tried both He initialization and the initialization recommended in Klambauer et al. (2017), and choose the best result for each model separately.

| Model | Top-1 Acc. (%) | Top-5 Acc. (%) |
|---|---|---|
| LReLU | 78.4 | 94.1 |
| PReLU | 77.7 | 93.5 |
| Softplus | 78.7 | 94.4 |
| ELU | 77.9 | 93.7 |
| SELU | 76.7 | 92.8 |
| GELU | 77.7 | 93.9 |
| ReLU | 78.4 | 94.2 |
| Swish-1 | 78.7 | 94.2 |
| Swish | 78.7 | 94.0 |

Table 9: Inception-v3 on ImageNet.

| Model | Top-1 Acc. (%) | Top-5 Acc. (%) |
|---|---|---|
| LReLU | 79.3 | 94.7 |
| PReLU | 79.3 | 94.4 |
| Softplus | 79.6 | 94.8 |
| ELU | 79.5 | 94.5 |
| SELU | 78.3 | 94.5 |
| GELU | 79.0 | 94.6 |
| ReLU | 79.2 | 94.6 |
| Swish-1 | 79.3 | 94.7 |
| Swish | 79.3 | 94.6 |

Table 10: Inception-v4 on ImageNet.

larger models, but performs worse on both mobile sized models. For Inception-v4, the gains from switching between activation functions is more limited, and Swish slightly underperforms Softplus and ELU. In general, the results suggest that switching to Swish improves performance with little additional tuning.

## 5.4 MACHINE TRANSLATION

We additionally benchmark Swish on the domain of machine translation. We train machine translation models on the standard WMT 2014 English→German dataset, which has 4.5 million training sentences, and evaluate on 4 different newstest sets using the standard BLEU metric. We use the attention based Transformer (Vaswani et al., 2017) model, which utilizes ReLUs in a 2-layered feed-forward network between each attention layer. We train a 12 layer "Base Transformer" model with 2 different learning rates[4] for 300K steps, but otherwise use the same hyperparameters as in the original work, such as using Adam (Kingma & Ba, 2015) to optimize.

| Model | newstest2013 | newstest2014 | newstest2015 | newstest2016 |
|---|---|---|---|---|
| LReLU | 26.2 | 27.9 | 29.8 | 33.4 |
| PReLU | 26.3 | 27.7 | 29.7 | 33.1 |
| Softplus | 23.4 | 23.6 | 25.8 | 29.2 |
| ELU | 24.6 | 25.1 | 27.7 | 32.5 |
| SELU | 23.7 | 23.5 | 25.9 | 30.5 |
| GELU | 25.9 | 27.3 | 29.5 | 33.1 |
| ReLU | 26.1 | 27.8 | 29.8 | 33.3 |
| Swish-1 | 26.2 | 28.0 | 30.1 | 34.0 |
| Swish | 26.5 | 27.6 | 30.0 | 33.1 |

Table 11: BLEU score of a 12 layer Transformer on WMT English→German.

Table 11 shows that Swish outperforms or matches the other baselines on machine translation. Swish-1 does especially well on newstest2016, exceeding the next best performing baseline by 0.6 BLEU points. The worst performing baseline function is Softplus, demonstrating inconsistency in performance across differing domains. In contrast, Swish consistently performs well across multiple domains.

## 6 RELATED WORK

Swish was found using a variety of automated search techniques. Search techniques have been utilized in other works to discover convolutional and recurrent architectures (Zoph & Le, 2016; Zoph et al., 2017; Real et al., 2017; Cai et al., 2017; Zhong et al., 2017) and optimizers (Bello et al., 2017). The use of search techniques to discover traditionally hand-designed components is an instance of the recently revived subfield of meta-learning (Schmidhuber, 1987; Naik & Mammone,

---

[4]We tried an additional learning rate for Softplus, but found it did not work well across all learning rates.

1992; Thrun & Pratt, 2012). Meta-learning has been used to find initializations for one-shot learning (Finn et al., 2017; Ravi & Larochelle, 2016), adaptable reinforcement learning (Wang et al., 2016; Duan et al., 2016), and generating model parameters (Ha et al., 2016). Meta-learning is powerful because the flexibility derived from the minimal assumptions encoded leads to empirically effective solutions. We take advantage of this property in order to find scalar activation functions, such as Swish, that have strong empirical performance.

While this work focuses on scalar activation functions, which transform one scalar to another scalar, there are many types of activation functions used in deep networks. *Many-to-one* functions, like max pooling, maxout (Goodfellow et al., 2013), and gating (Hochreiter & Schmidhuber, 1997; Srivastava et al., 2015; van den Oord et al., 2016; Dauphin et al., 2016; Wu et al., 2016; Miech et al., 2017), derive their power from combining multiple sources in a nonlinear way. *One-to-many* functions, like Concatenated ReLU (Shang et al., 2016), improve performance by applying multiple nonlinear functions to a single input. Finally, *many-to-many* functions, such as BatchNorm (Ioffe & Szegedy, 2015) and LayerNorm (Ba et al., 2016), induce powerful nonlinear relationships between their inputs.

Most prior work has focused on proposing new activation functions (Maas et al., 2013; Agostinelli et al., 2014; He et al., 2015; Clevert et al., 2015; Hendrycks & Gimpel, 2016; Klambauer et al., 2017; Qiu & Cai, 2017; Zhou et al., 2017; Elfwing et al., 2017), but few studies, such as Xu et al. (2015), have systematically compared different activation functions. To the best of our knowledge, this is the first study to compare scalar activation functions across multiple challenging datasets.

Our study shows that Swish consistently outperforms ReLU on deep models. The strong performance of Swish challenges conventional wisdom about ReLU. Hypotheses about the importance of the gradient preserving property of ReLU seem unnecessary when residual connections (He et al., 2016a) enable the optimization of very deep networks. A similar insight can be found in the fully attentional Transformer (Vaswani et al., 2017), where the intricately constructed LSTM cell (Hochreiter & Schmidhuber, 1997) is no longer necessary when constant-length attentional connections are used. Architectural improvements lessen the need for individual components to preserve gradients.

## 7 CONCLUSION

In this work, we utilized automatic search techniques to discover novel activation functions that have strong empirical performance. We then empirically validated the best discovered activation function, which we call Swish and is defined as $f(x) = x \cdot \text{sigmoid}(\beta x)$. Our experiments used models and hyperparameters that were designed for ReLU and just replaced the ReLU activation function with Swish; even this simple, suboptimal procedure resulted in Swish consistently outperforming ReLU and other activation functions. We expect additional gains to be made when these models and hyperparameters are specifically designed with Swish in mind. The simplicity of Swish and its similarity to ReLU means that replacing ReLUs in any network is just a simple one line code change.

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
