# OpenReview forum: "Searching for Activation Functions"
_ICLR.cc/2018/Conference — Invite to Workshop Track_

### Official Review · AnonReviewer3 · 2017-11-24
**Another approach for arriving at proven concepts on activation functions**

**Rating:** 4
**Confidence:** 4

**Review:**

Authors propose a reinforcement learning based approach for finding a non-linearity by searching through combinations from a set of unary and binary operators. The best one found is termed Swish unit; x * sigmoid(b*x).

The properties of Swish like allowing information flow on the negative side and linear nature on the positive have been proven to be important for better optimization in the past by other functions like LReLU, PLReLU etc. As pointed out by the authors themselves for b=1 Swish is equivalent to SiL proposed in Elfwing et. al. (2017).

In terms of experimental validation, in most cases the increase is performance when using Swish as compared to other models are very small fractions. Again, the authors do state that "our results may not be directly comparable to the results in the corresponding works due to differences in our training steps."

Based on the Figure 6 authors claim that the non-monotonic bump of Swish on the negative side is very important aspect. More explanation is required on why is it important and how does it help optimization. Distribution of learned b in Swish for different layers of a network can interesting to observe.

---

> ### Author Response · Authors · 2018-01-01
> **Re: Reviewer3**
>
> We don’t completely understand the reviewer’s rationale for rejection. Is it because of the lack of novelty, the inconsistent gains, or the work being insignificant?
>
> First, in terms of the work being significant, we want to emphasize that ReLU is the cornerstone of deep learning models. Being able to replace ReLU is extremely impactful because it produces a gain across a large number of models. So in terms of impact, we believe that our work is significant.
>
> Secondly, in terms of inconsistent gains, the signed tests already confirm that the gains are statistically significant in our experiments. These results suggest that switching to Swish is an easy and consistent way of getting an improvement regardless of which baseline activation function is used. Unlike previous studies, the baselines in our work are extremely strong: they are state-of-the-art models where the models are built with ReLUs as the default activation. Furthermore, the same amount of tuning was used for every activation function, and in fact, many non-Swish activation functions actually got more tuning. Thus, it is unreasonable to expect a huge improvement. That said, in some cases, Swish on Imagenet makes a more than 1% top-1 improvement. For context, the gap between Inception-v3 and Inception-v4 (a year of work) is only 1.2%.
>
> Finally, in terms of novelty, our work differs from Elfwing et al. (2017) in a number of significant ways. They just propose a single activation function, whereas our work searches over a vast space of activation functions to find the best empirically performing activation function. The search component is important because we save researchers from the painful process of manually trying out a number of individual activation functions in order to find one that outperforms ReLU (i.e., graduate student descent). The activation function found by this search, Swish, is more general than the other proposed by Elfwing et al. (2017). Another key contribution is our thorough empirical study. Their activation function was tested only on relatively shallow reinforcement learning models. We performed a thorough experimental evaluation on many challenging, deep, large-scale supervised models with extremely strong baselines. We believe these differences are significant enough to differentiate us.
>
> The non-monotonic bump, which is controlled by beta, has gradients for negative preactivations (unlike ReLU). We have plotted the beta distribution over the each layer Swish here: https://imgur.com/a/AIbS2 . Note this is on the Mobile NASNet-A model, which has many layers composed in parallel (similar to Inception and unlike ResNet). The plot suggests that the tuneable beta is flexibly used. Early layers use large values of beta, which corresponds to ReLU-like behavior, whereas later layers tend to stay around the [0, 1.5] range, corresponding to a more linear-like behavior.

---

> > ### Comment · AnonReviewer3 · 2018-01-03
> > **Reply:**
> >
> > Yes, I do agree that ReLU is one of the major reason for improvement of deep learning models. But, it is not just because ReLU was able to experimentally beat performance of existing  non-linearities by a small fraction.
> >
> > The fractional increase in performance on benchmarks can be because of various reasons, not just switching non-linearity. For example, in many cases a simple larger batch size can result in small fractional change in performance. The hyper-parameter settings in which other non-linearities might perform better can be different than the ones more suitable for proposed non-linearity. Also, I do not agree that the search factor helps researchers to save time on trying out different non-linearities, still one has to spend time on searching best 'betas' (which will result in small improvement over benchmarks) for every dataset. I would rather use a more well understood non-linearity which gives reasonable results on benchmarks.
> >
> > The properties of the non-linearities proposed in the article like "allowing information flow on the negative side and linear nature on the positive  side"(also mentioned in my review) have been proven to be important for better optimization in the past by other functions like LReLU, PLReLU etc.
> >
> > The results from the article show that Swish-1 ( or SiL from Elfwing et al. (2017)) performs same as Swish.

---

> > > ### Author Response · Authors · 2018-01-06
> > > **Re: Reviewer3**
> > >
> > > Thank you for the comment.
> > >
> > > [[Our activation only beats other nonlinearities by “a small fraction”]] First of all, we question the conventional wisdom that ReLU greatly outperforms tanh or sigmoid units in modern architectures. While AlexNet may benefit from the optimization properties of ReLU, modern architectures use BatchNorm, which eases optimization even for sigmoid and tanh units. The BatchNorm paper [1] reports around a 3% gap between sigmoid and ReLU (it’s unclear if the sigmoid experiment was with tuning and this experiment is done on the older Inception-v1). The PReLU paper [2], cited 1800 times, proposes PReLU and reports a gain of 1.2% (Figure 3), again on a much weaker baseline. We cannot find any evidence in recent work that suggests that gap between sigmoid / tanh units and ReLU is huge. The gains produced by Swish are around 1% on top of much harder baselines, such as Inception-ResNet-v2, is already a third of the gain produced by ReLU and on par with the gains produced by PReLU.
> > >
> > > [[Small fraction gained due to hyperparameter tuning]] We want to emphasize how hard it is to get improvements on these state-of-art models. The models we tried (e.g., Inception-ResNet-v2) have been **heavily tuned** using ReLUs. The fact that Swish improves on these heavily tuned models with very minor additional tuning is impressive. This result suggests that models can simply replace the ReLUs with Swish units and enjoy performance gains. We believe the drop-in-replacement property of Swish is extremely powerful because one of the key impediments to the adoption of a new technique is the need to run many additional experiments (e,g,, a lot of hyperparameter tuning).  This achievement is impactful because it enables the replacement of ReLUs that are widely used across research and industry.
> > >
> > > [[Searching for betas]] The reviewer also misunderstands the betas in Swish. When we use Swish-beta, one does not need to search for the optimal value of beta because it can be learned by backpropagation.
> > >
> > > [[Gradient on the negative side]] We do not claim that Swish is the first activation function to utilize gradients in the negative preactivation regime. We simply suggested that Swish may benefit from same properties utilized by LReLU and PReLU.
> > >
> > > [1] Sergey Ioffe, Christian Szegedy. Batch Normalization: Accelerating Deep Network Training by Reducing Internal Covariate Shift. In JMLR, 2015. (See Figure 3: https://arxiv.org/pdf/1502.03167.pdf )
> > > [2] Kaiming He, Xiangyu Zhang, Shaoqing Ren, Jian Sun. Delving Deep into Rectifiers: Surpassing Human-Level Performance on ImageNet Classification. In CVPR, 2015 (See Table 2: https://arxiv.org/pdf/1502.01852.pdf )

---

### Official Review · AnonReviewer1 · 2017-11-27

**Rating:** 5
**Confidence:** 5

**Review:**

This paper is utilizing reinforcement learning to search new activation function. The search space is combination of a set of unary and binary functions. The search result is a new activation function named Swish function. The authors also run a number of ImageNet experiments, and one NTM experiment.

Comments:

1. The search function set and method is not novel.
2. There is no theoretical depth in the searched activation about why it is better.
3. For leaky ReLU, use larger alpha will lead better result, eg, alpha = 0.3 or 0.5. I suggest to add experiment to leak ReLU with larger alpha. This result has been shown in previous work.

Overall, I think this paper is not meeting ICLR novelty standard. I recommend to submit this paper to ICLR workshop track.

---

> ### Author Response · Authors · 2018-01-01
> **Re: Reviewer1**
>
> 1. Can the reviewer explain further why our work is not novel? Our activation function and the method to find it have not been explored before, and our work holds the promise of improving representation learning across many models.  Furthermore, no previous work has come close to our level of thorough empirical evaluation. This type of contribution is as important as novelty -- it can be argued that the resurgence of CNNs is primarily due to conceptually simple empirical studies demonstrating their effectiveness on new datasets.
>
> 2. We respectfully disagree with the reviewer that theoretical depth is necessary to be accepted. Following this argument, we can also argue that many extremely useful techniques in representation / deep learning, such as word2vec, ReLU, BatchNorm, etc, should not be accepted to ICLR because the original papers did not supply theoretical results about why they worked. Our community has typically followed that paradigm of discovering techniques experimentally and further work analyzing the technique. We believe our thorough and fair empirical evaluation provides a solid foundation for further work analyzing the theoretical properties of Swish.
>
> 3. We experimented with the leaky ReLU using alpha = 0.5 on Inception-ResNet-v2 using the same hyperparameter sweep, and and did not find any improvement over the alpha used in our work (which was suggested by the original paper that proposed leaky ReLUs).

---

> > ### Comment · AnonReviewer1 · 2018-01-12
> > **Reply**
> >
> > 1. Novelty
> >
> > The methodology of searching has been used in Genetic Programming for a long time. The RNN controller has been used in many paper from Google Brain. This paper's contribution is using RL to search in a GP flavor. Although it is new in activation function search field, in methodology view, it is not novel.
> >
> > 2. Theoretical depth
> >
> > Actually, BatchNorm and ReLU provides its explanation of why they work in the original paper and the explanation was accepted by community for a long time. I understand how deep learning community's experimentally flavor, but activation function is a fundamentally problem in understanding how neural network works. Swish performs similarly or slightly better compare to the commonly used activation functions. If without any theoretical explanation, it is hard to acknowledge it as a breaking research. What's more, different activation function may requires different initialization and learning rate, I respect the authors have enough computation power to sweep, but without any theoretical explanation, the paper is more like a experiment report rather than a good ICLR paper.

---

### Official Review · AnonReviewer4 · 2017-12-06
**Well written paper and well conducted experiments.**

**Rating:** 7
**Confidence:** 5

**Review:**

The author uses reinforcement learning to find new potential activation functions from a rich set of possible candidates. The search is performed by maximizing the validation performance on CIFAR-10 for a given network architecture. One candidate stood out and is thoroughly analyze in the reste of the paper. The analysis is conducted across images datasets and one translation dataset on different architectures and numerous baselines, including recent ones such as SELU. The improvement is marginal compared to some baselines but systematic. Signed test shows that the improvement is statistically significant.

Overall the paper is well written and the lack of theoretical grounding is compensated by a reliable and thorough benchmark. While a new activation function is not exiting, improving basic building blocks is still important for the community.

Since the paper is fairly experimental, providing code for reproducibility would be appreciated.

---

> ### Author Response · Authors · 2018-01-01
> **Re: Reviewer4**
>
> The reviewer suggested “Since the paper is fairly experimental, providing code for reproducibility would be appreciated”. We agree, and we will open source some of the experiments around the time of acceptance.

---

### Public Comment · (anonymous) · 2017-11-04
**non-monotonic vs. small negative negative for negative pre-activations**

You state: "In Figure 6, a large percentage of preactivations fall inside the domain of the bump (−5 ≤ x ≤ 0), which indicates that the non-monotonic bump is an important aspect of Swish."

It seems that non-monotonic behavior is an artifact of your function that could have negative consequences by making a "bumpier" loss surface for optimizers. What is the value of Swish approaching 0 as x heads to -inf? Why wouldn't small negative values be sufficient for all negative pre-actiations (x ≤ -5)?

Wouldn't something like CELU with small alpha in the long run be better?  CELU paper:
https://arxiv.org/pdf/1704.07483.pdf

---

### Public Comment · (anonymous) · 2017-11-16
**Related work**

You mention this in the body, but it would be helpful in the related work if you pointed out that (Hendrycks & Gimpel, 2016) considered this activation function but found a slightly different version to be better, and that Elfwing et. al already proposed Swish-1 under a different name.

I see you went from sigmoid(x) -> sigmoid(beta * x) to avoid outright duplication, but empirically it looks like Swish-1 is equal or better than Swish?

Table 3 is a little misleading - the magnitude of the differences is what we really care about, and those magnitudes are quite small.

Figure 8 is a little misleading - ReLU's are far and away the worst on that particular dataset+model, I imagine the plot for existing work like PReLU, which gives basically the same performance, would look very different.

In the original version, you bolded the non-ReLU activations which provide basically the same perf, but you don't in the new version - why not? PReLU is often the same as Swish, but without the bolding it's a lot harder to read.

The differences in perf are small enough to make me think this is just hyperparameter noise. For instance, you try 2 learning rates for the NMT results, why only 2? What 2 did you choose? Why did you choose them? If you had introduced PReLU, would it's numbers be higher? Concrete questions aside, I have a very hard time trusting this paper.

---

> ### Public Comment · (anonymous) · 2018-01-07
> **No response from authors**
>
> The authors appear to have made a decision to ignore all comments which are not from reviewers. To be clear, if I were a reviewer, I would score this paper as a 4 with confidence of 4.
>
> In addition to the above issues, I'd point out that ReLU isn't the only baseline here - to claim a worthwhile contribution, they also need to demonstrate improvement over functions such as PReLU, where the empirical evidence is even weaker to non-existent.

---

### Public Comment · (anonymous) · 2017-11-16
**Figure 7 would be more helpful if more typical beta values were shown**

Given the distribution of actual learned β values for Swish the were presented in Figure 7, it would be more instructive to show β=0, β=0.3, β=0.5, β=1.0 in Figures 4&5. While β=10.0 is interesting to look at in the 1st derivative plot, it doesn’t seem to have been learned as useful value for β.

---

### Public Comment · (anonymous) · 2017-11-16
**Figure 8 should show PReLU given data in Table 6**

Figure 8 plot should show PReLU not ReLU since given data in Table 6, PReLU is better than ReLU in every case.

in addition, in many of the other results in the paper LReLU is slightly better than  PReLU.  The two differences are that LReLU has α=0.01 and PReLU at α=.25 and that α in PReLU is learnable. Looking closely at Swish and PReLU plots, a more comparable starting initialization for PReLU would be α=.10 and it would be somewhat closer to the value the you use for LReLU.

We suggest rerunning PReLU with α=.10 and putting this result in Figure 8 and Table 6.

---

### Public Comment · (anonymous) · 2017-11-16
**Insights from Learnable Swish parameter(β)**

Figure 7 shows an interesting feature that the β=1 is the most prevalent single β value after training.  Since Swish smoothly varies with β, one can only assume that the reason for this inconsistency was that β was initialized to 1 and that during training this parameter was not adjusted in many cases.  The text of the paper should clearly state the initialization value of β.

The more interesting aspect of this distribution is that over 2x more β values were learned to be better in the range of (0.0 to 0.9) than at the (assumed) starting value of β=1.  β’s in this range suggests that larger negative values must have some advantage.

It would be very interesting to see understand if distribution of β values changes in the different layers of the neural network. Are the β in the range (0.0 to 0.9) more important at higher levels or lower levels.  It would also be instructive to see the effects of starting with β at another initial starting value.

Swish approaches x/2 as β approaches inf, why is this better than approaching x in the manner that PReLU does?

While the paper asserts the non-monotonic feature of Swish as an important aspect of Swish, but there is nothing that explains why this could be an advantage. In fact for Figure 6 show most negative preactivations are between -6 and 0 and given that Figure 7 shows most β between 0 and 1 most negative values will not be effected by non-monotonic behavior. Might the real lesson of the paper be that a smooth activation function with a smooth and continuous derivative function with a "learnable" small domain of negative values is more important for learning and generalization than non-montonicity?

---

### Author Response · Authors · 2018-01-06
**Clearing up concerns and misunderstandings**

We thank the reviewers for their comments and feedback. We are extremely surprised by the low scores for the paper that proposes a novel method that finds better activation functions, one of which has a potential to be better than ReLUs. During the discussion with the reviewers, we have found a few major concerns and misunderstandings amongst the reviewers, and we want to bring it up to a general discussion:

The reviewers are concerned that our activation only beats other nonlinearities by “a small fraction”. First of all, we question the conventional wisdom that ReLU greatly outperforms tanh or sigmoid units in modern architectures. While AlexNet may benefit from the optimization properties of ReLU, modern architectures use BatchNorm, which eases optimization even for sigmoid and tanh units. The BatchNorm paper [1] reports around a 3% gap between sigmoid and ReLU (it’s unclear if the sigmoid experiment was with tuning and this experiment is done on the older Inception-v1). The PReLU paper [2], cited 1800 times, proposes PReLU and reports a gain of 1.2%, again on a much weaker baseline. We cannot find any evidence in recent work that suggests that gap between sigmoid / tanh units and ReLU is huge. The gains produced by Swish are around 1% on top of much harder baselines, such as Inception-ResNet-v2, is already a third of the gain produced by ReLU and on par with the gains produced by PReLU.

The reviewers are concerned that the small gains are simply due to hyperparameter tuning. We stress here that unlike many prior works, the models we tried (e.g., Inception-ResNet-v2) have been **heavily tuned** using ReLUs. The fact that Swish improves on these heavily tuned models with very minor additional tuning is impressive. This result suggests that models can simply replace the ReLUs with Swish units and enjoy performance gains. We believe the drop-in-replacement property of Swish is extremely powerful because one of the key impediments to the adoption of a new technique is the need to run many additional experiments (e,g,, a lot of hyperparameter tuning).  This achievement is impactful because it enables the replacement of ReLUs that are widely used across research and industry.

The reviewers are also concerned that our activation function is too similar to the work by Elfwing et al. When we conducted our research, we were honestly not aware of the work by Elfwing et al (their paper was first posted fairly recently on arxiv in Feb, 2017 and to the best of our knowledge, not accepted to any mainstream conference). That said, we have happily cited their work and credited their contributions. We are also happy to reuse the name “SiL” proposed by Elfwing et al if the reviewers see fit. In that case, Elfwing et al should be thrilled to know that their proposal is validated through a thorough search procedure. We also want to emphasize a number of key differences between our work and Elfwing et al. First, the focus of our paper is to search for an activation functions. Any researcher can use our recipes to drop in new primitives to search for better activation functions. Furthermore, our work has much more comprehensive empirical validation. Elfwing et al. only conducted experiments on relatively shallow reinforcement learning tasks, whereas we evaluated on challenging supervised benchmarks such as ImageNet with extremely tough baselines and equal amounts of tuning for fairness. We believe that we have conducted the most thorough evaluation of activation functions among any published work.

Please reconsider your rejection decisions.

[1] Sergey Ioffe, Christian Szegedy. Batch Normalization: Accelerating Deep Network Training by Reducing Internal Covariate Shift. In ICML, 2015. (See Figure 3: https://arxiv.org/pdf/1502.03167.pdf )
[2] Kaiming He, Xiangyu Zhang, Shaoqing Ren, Jian Sun. Delving Deep into Rectifiers: Surpassing Human-Level Performance on ImageNet Classification. In CVPR, 2015 (See Table 2: https://arxiv.org/pdf/1502.01852.pdf )

---

### Decision · Program_Chairs · 2018-01-29
**ICLR 2018 Conference Acceptance Decision**

**Decision:**

Invite to Workshop Track

**Comment:**

The author's propose to use swish and show that it performs significantly better than Relus on sota vision models. Reviewers and anonymous ones counter that PRelus should be doing quite well too. Unfortunately, the paper falls in the category where it is hard to prove the utility of the method through one paper alone, and broader consensus relies on reproduction by the community. As a results, I'm going to recommend publishing to a workshop for now.